# Occurrence and Characteristics of *Staphylococcus aureus* Strains along the Production Chain of Raw Milk Cheeses in Poland

**DOI:** 10.3390/molecules27196569

**Published:** 2022-10-04

**Authors:** Joanna Gajewska, Wioleta Chajęcka-Wierzchowska, Anna Zadernowska

**Affiliations:** Department of Industrial and Food Microbiology, Faculty of Food Science, University of Warmia and Mazury in Olsztyn, Plac Cieszyński 1, 10-726 Olsztyn, Poland

**Keywords:** *S. aureus*, staphylococcal enterotoxins genes, antimicrobial resistance, virulence, raw milk cheese

## Abstract

Cheeses produced from unpasteurized milk by traditional production methods may contain many groups of microorganisms, including *Staphylococcus aureus.* The aim of this study was to determine the occurrence of *S. aureus* in the artisanal cheese production chain from unpasteurized milk. We investigated the prevalence of *S. aureus* strains isolated from various stages of artisanal cheese of unpasteurized milk production from farms in the northeastern and southern parts of Poland and characterized them. Characterization included antimicrobial susceptibility by microbroth dilution and biofilm formation by in vitro assay. Among all strains, the presence of enterotoxigenic genes and genes involved with biofilm formation and antibiotic resistance were screened by PCR-based methods. A total of 180 samples were examined. A high percentage of strains were resistant to penicillin (54/58.1%) and tobramycin (32/34.4%). Some tested isolates also showed resistance to the macrolide class of antibiotics: azithromycin, clarithromycin, and erythromycin at 17/18.3%, 15/16.1%, and 21/22.6%, respectively. Among tested isolates, we also found phenotypic resistance to oxacillin (9/9.7%) and cefoxitin (12/12.9%). The *bla*Z gene encoding penicillin resistance was the most common gene encoding antibiotic resistance among the tested strains. All isolates showing phenotypic resistance to cefoxitin possessed the *mec*A gene. The study also evaluated the prevalence of biofilm-associated genes, with *eno* the most frequently associated gene. Eighty-nine out of 93 *S. aureus* isolates (95.7%) possessed at least one enterotoxin-encoding gene. The results of this study showed that production of raw milk cheeses may be a source of antibiotic resistance and virulent *S. aureus*. Our results suggest that artisanal cheese producers should better control production hygiene.

## 1. Introduction

The consumption of milk and dairy products has a long tradition in human nutrition [1]. Milk production is one of the most important branches of the agricultural economy in Poland, and each year production increases. Milk and dairy products have been shown to have a positive impact on human health due to the abundance of nutrients present within them, such as calcium, potassium, proteins, fat and vitamins [2]. In recent years, traditional foods such as farmhouse artisanal cheese have become increasingly popular with consumers in Poland [3]; an upward trend in the consumption of raw milk and raw milk processed products purchased in local markets and/or farms has been observed.

Cheeses produced from unpasteurized milk by traditional production methods may contain very diverse microbiota, which impacts their unique organoleptic characteristics [4]. However, due to their rich source of chemical compounds, these cheeses are also an excellent environment for the development of many groups of microorganisms, including *S. aureus* [5]. 

The occurrence of *S. aureus* in an artisanal cheese production chain may be due to the fact that milk is subjected to low-temperature processing without the pasteurization process, which creates the risk of the occurrence of *S. aureus* in finished products. Furthermore, artisanal production involves the risk of contamination of finished products at various stages of production [6] due to unhygienic practices. In addition to low milk quality, each further processing step facilitates the entrance of *S. aureus* in the dairy chain. One possible source of contamination of finished products is the people involved in cheese manufacturing, since *S. aureus* is frequently present on the skin of cheesemakers [4]. Along with this human-to-food contamination route, several other entrance points of *S. aureus* into the dairy chain have been described previously [7]. One factor allowing *S. aureus* to grow during cheese production is insufficient acidification by the starter cultures of the lactic acid bacteria. In addition, biofilm formation on dairy equipment allows niches to grow and contaminate the processing line [7,8]. 

*S. aureus* is one of the most important causative agents of food poisoning worldwide. *S. aureus* strains are known to be resistant to several types of antibiotics [9]. Overuse of antibiotics causes selective pressure in animals and humans to develop antibiotic resistance [10]. Nowadays, the most severe types of antibiotic-resistant *S. aureus* are methicillin-resistant *S. aureus* (MRSA) and vancomycin-resistant *S. aureus* (VRSA) [11]. MRSA strains are of great importance in public health and can pose a zoonotic risk to humans. *S. aureus* can be transmitted by contaminated food and cause food-borne outbreaks, which are mostly caused by toxins. On the other hand, *S. aureus* can also lead to severe diseases, e.g., sepsis, endocarditis, and necrotizing pneumonia [12]. Methicillin resistance of *S. aureus*, important in farming and food-processing plants, stems from the possibility of zoonotic infection of consumers and workers involved in animal husbandry [13]. Global surveillance has shown that MRSA is a problem in some continents and countries where studies have been carried out, increasing mortality and the need to use expensive last-resource antibiotics [14,15,16,17]. MRSA strains are those that carry the *mec*A and/or *mec*C gene and are resistant to all penicillins, cephalosporins, and carbapenem [18]. Vancomycin is an important antibacterial agent used to treat serious infections caused by MRSA [19]. It is worth noting that, at present, the emergence and spread of VRSA strains remains a challenge to the global health crisis due to the lack of effective control and treatment efficacy [20].

Some *S. aureus* isolates are capable of producing enterotoxins, which are potent emetic agents. Staphylococcal food poisoning (SFP) is caused by the consumption of products containing enterotoxins known as superantigens (SAgs) [4]. Consumption of products containing enterotoxins may cause a diverse range of gastrointestinal symptoms, including nausea, violent vomiting, and abdominal cramping, with or without diarrhea [21]. The ability to form biofilm among *S. aureus* strains is also known. In the food industry, *S. aureus* biofilm may be distributed throughout the dairy food chain by the food raw products, raw milk tanks, and the food processing environment and equipment [22]. Microbial biofilms that form on the surfaces of machinery and equipment in food processing plants affect not only the safety and quality of the final product, but also the technological process flow [23]. This ability is mediated by many genetic determinants, including operon *ica*ADBC, which encodes the synthesis of the polysaccharide intercellular adhesin (PIA) molecule, the *agr* locus, and different genes recognized as microbial surface components recognizing adhesive matrix molecules (MSCRAMMs) [22]. 

Nowadays, studies on the prevalence and characterization of *S. aureus* isolated from final dairy products available for consumers are very common, with examples from China, Norway, and Greece, among other countries [24,25,26,27,28,29]. In addition, numerous articles have been published on the prevalence and characterization of *S. aureus* in raw milk, especially in China, Iran, and Australia [30,31,32,33,34]. It is worth emphasizing that *S. aureus* may enter the food chain during processing and preparation of the products [35]. Despite this, few data are available on the prevalence and characteristics of *S. aureus* in the food production chain, especially in Poland [4,36]. 

Therefore, the aim of the study was to determine the occurrence of *S. aureus* in an artisanal cheese production chain from unpasteurized milk. Moreover, the antibiotic resistance, enterotoxicity, and biofilm formation and slime production, as well as the frequency of virulence-related genes among *S. aureus* strains were investigated. 

## 2. Results

### 2.1. Occurrence of S. aureus Strains along the Cheese Production Chain 

In the current study, the presence of *S. aureus* was found in 93/180 (51.7%) of all samples tested for *S. aureus* (Table 1). *S. aureus* was isolated from raw material, semi-products, and in the final product. *S. aureus* was also found on food contact surfaces—form (9/50.0%), tank (8/44.4%), and jar (8/44.4%)—and in a small percentage of non-food contact surfaces: sinks (2/11.1%).

### 2.2. Antibiotic Resistance Profiles among S. aureus Strains

In this study, all the *S. aureus* (n = 93) isolates were evaluated for the phenotypic resistance to antibiotics and the presence antibiotic-resistant genes.

Resistance to benzypenicillin was the most common among the tested *S. aureus* strains (54/93 (58.1%)), followed by tobramycin (32/93 (34.4%)). Isolates were also resistant to macrolides: azithromycin, clarithromycin, and erythromycin: 17/93 (18.3%), 15/93 (16.1%), and 21/93 (22.6%), respectively. Resistance to tetracycline and tigecycline belonging to the tetracycline class was shown among 19/93 (20.4%) and 22/93 (23.7%) of isolates, respectively. Among tested isolates, phenotypic resistance to oxacillin (9/93 (9.7%)) and cefoxitin (12/93 (12.9%)) were also shown both in raw materials and in ready-to-eat cheeses. The isolates’ resistance to the other tested antibiotics ranged from 1.1% to 16.1%, with none of the isolates being resistant to gentamycin, clindamycin, rifampicin, moxifloxacin, or chloramphenicol (Table 2). Multidrug-resistant (MDR) status of *S. aureus* isolates was tested against 15 classes of antimicrobials. Multi-drug resistance in this study was taken as resistant to at least three antibiotics from different chemical classes of antibiotic. Therefore, among all isolates of *S. aureus*, 21/93 (22.6%) were defined as multidrug-resistant. It was found that the multiple antibiotic resistance index (MAR) among isolates ranged from 0.04 to 0.43, with an overall mean of 0.12. In this study, 12/93 (12.9%) isolates were defined as MRSA by the cefoxitin/oxacillin MIC test and were positive for the *mec*A gene, and two strains 2/93 (2.2%) were classified as phenotypic VRSA based only on phenotypic results using microdilution assay.

The presence of tested resistance genes ranged from 12/93 (12.9%) to 89/93 (95.7%) among all *S. aureus* isolates. The most prevalent gene encoding antibiotic resistance among the tested isolates was *bla*Z (89/93 (95.7%)), encoding penicillinase. All isolates showing phenotypic resistance to cefoxitin possessed the *mec*A gene (12/93 (12.9%)). Although a small percentage of strains showed phenotypic resistance to tetracycline, a high percentage of strains possessing the *tet*(K), *tet*(L), and *tet*(M) genes were found (59/93 (63.4%), 16/93 (17.2%), and 76/93 (81.7%), respectively). The presence of *aac(6′)-Ie-aph(2″)-Ia* and *aph(3′)IIIa* genes, which determine aminoglycoside resistance, was also found among the tested isolates, and were present among 56/93 (60.2%) and 86/93 (92.5%) of the isolates, respectively. The tested strains also harbored an *erm*B gene in 27/93 (29.0%) (Table 3).

Statistical analysis showed a positive but not significant at *p* < 0.05 correlation between occurrence of the *erm*(B) gene and phenotypically resistant clarithromycin (r = 0.2348), azithromycin (*r* = 0.4943), and erythromycin (*r* = 0.5041) (*p* < 0.05). Further, we found positive correlation between *tet*(K) and phenotypic resistance to tetracycline (*r* = 0.163) and tigecycline (*r* = 0.265) (*p* < 0.05) and negative but not significant at *p* < 0.05 correlation between *tet*(L) and phenotypic resistance to tetracycline among isolates (*r* = −0.090).

### 2.3. Presence of Virulence Factors in S. aureus Isolates 

Among all isolates, including those from raw materials and swabs from surfaces, the ability to form biofilm has been found. Biofilm formation by *S. aureus* included the following: weak formers 33 (35.4%), intermediate formers 2 (2.2%), and strong formers 58 (62.4%). Table 4 summarizes the biofilm-forming strength of tested *S. aureus* isolates at different stages of cheese production. There was no association between slime production and the ability to form biofilm at *p* < 0.05 (*p* = 0. 0981026).

The study also evaluated the prevalence of biofilm-associated genes, and it was shown that the most frequent gene was *eno* (laminin-binding protein), which was observed among all isolates. Moreover, *bap* (biofilm-associated protein) was found among 32/34.4% of all isolates. The genes *sig*B, *sar*A, and *agr*D were found in 42/45.2%, 74/79.6%, and 33/35.5% of isolates, respectively. After analyzing the relationship between virulence genes and biofilm formation, it was shown that there were significant correlations between the occurrence of *agr*D gene and biofilm formation (*r* = 0.3038). Among all isolates, we found a positive but not significant correlation between the occurrence of *agr*D gene and *sar*A (*r* = 0.0414) and *sig*B and *sar*A (*r* = 0.4055) (*p* < 0.05).

Each strain was tested for the presence of staphylococcal enterotoxins genes. Eighty-nine out of 93 *S. aureus* isolates (95.7%) possessed at least one enterotoxin-encoding gene. It was observed that genes encoding SEI (enterotoxin-like toxin) were more frequent than genes encoding classical enterotoxins (SEs) among the isolates studied. Among the genes encoding SEI, the presence of *selk* (61.3%) was recorded most frequently, followed by *selg* (58.1%), *seg* (46.2%), *seln* (45.2%), *selu* (31.2%), *selm* (19.4%), and *selj* (28%). It is worth noting that the highest percentage of the presence of enterotoxigenic genes was found among strains isolated from finished products. Statistical analysis showed positive correlation between co-occurrence of some enterotoxins genes, e.g., *selp* and *selk* (*r* = 0.0698), *selp* and *selu* (*r* = 0.1061), *selm* and *selk* (*r* = 0.1709), *selm* and *selo* (*r* = 0.8582), and *selm* and *seg* (*r* = 0.3676).

Statistical analysis showed no statistically significant differences between the source of isolation of *S. aureus* and the ability to produce slime (*p* > 0.999), resistance to antimicrobial agents (*p* = 0.4703), or occurrence of virulence genes (*p* > 0.999).

To systematically detect all such associations in an unbiased way, we analyzed all potential pairs of variables (except those that were entirely present or completely absent among all (n = 93) isolates. Among the observed correlations, we found positive correlations between the ability of the strains to form biofilm and phenotypic resistance to all the antibiotics tested. Interestingly, we found negative correlations between slime production and phenotypic resistance to trimethoprim, kanamycin, and quinupristin/dalfopristin (Figure 1).

## 3. Discussion

In the current study, the occurrence, antibiotic resistance, and virulence of *S. aureus* isolated at different production stages of raw milk cheeses were tested. There are currently few studies focusing on assessing the whole production chain of cheeses from unpasteurized milk [4,37,38]. Most studies have focused on the evaluation of *S. aureus* strains isolated from raw milk from cows with clinical or subclinical mastitis or only in finished products [24,29,39,40,41,42]. Nevertheless, estimation of the prevalence and genetic determinants of *S. aureus* is always important to facilitate the implementation of rational mitigation strategies and to avoid the dissemination of this pathogen through the food chain [43].

Compared to the results obtained for the presence of *S. aureus* in unpasteurized milk cheeses (finished products), previous studies reported percentages of samples in which *S. aureus* was found as follows: 56.4% (44/78) in Mexico [44], 87.3% (62/71) in Serbia [45], and 45% (31/69) in Norway [46]. The different results regarding the prevalence of *S. aureus* in different countries may be related to differences in cheese production worldwide. It is assumed that the presence of *S. aureus* in farmstead cheeses may be due to the low quality of raw material, resulting from poor hygienic conditions during production and inadequate storage conditions, unclean hands of workers, and/or failure to follow basic GMP and GHP principles.

In this study, we observed that among the tested *S. aureus* strains, resistance to β-lactams (which include penicillin) was the most frequent. These observations are in line with reports by other authors, who have reported in recent years on the worldwide increase in *S. aureus* resistance to β-lactam antibiotics [9,41,47,48]. The high percentage of penicillin-resistant strains may be due to the frequency of use of this antibiotic in the treatment of cattle infections in Poland [4]. 

Tetracyclines are also commonly used to treat infections in cattle [49]. In the present study, a lower incidence of tetracycline resistance was observed (19/93 (20.4%)) compared to studies by other authors, which showed resistance to this antibiotic in *S. aureus* strains ranging from 19–100% [50,51]. Another antibiotic against which the tested strains showed resistance was erythromycin (21/93 (22.6%)), which is a macrolide antibiotic. Macrolides are commonly used to treat mastitis in cattle. Erythromycin is also used to treat staphylococcal infections in patients allergic to penicillin [41]. 

Nevertheless, compared to the present results, studies by other authors have shown a higher percentage of strains resistant to this antibiotic (22.4% and 44%) [41,52]. The *erm* genes confer cross-resistance to macrolides, lincosamides, and streptogramins B [53]. Antimicrobial resistance results obtained in this study correlate with antibiotics that are used to treat cattle infections in Poland. According to a report by the European Medicines Agency, in Poland in 2020, tetracyclines, penicillins, and macrolides represented over 70% of total veterinary antimicrobial agents sold [54]. 

The study showed that a small percentage (21/93 (22.6%)) of the tested isolates were multidrug-resistant (resistant to at least three antibiotics from different chemical classes of antibiotic), with 19.4% of isolates having a MAR index of >0.20. Nevertheless, it is worth emphasizing the need for constant prevention of excessive use of antibiotics in animal treatment, and it is necessary to monitor the prevalence of antibiotic-resistant foodborne pathogens, including *S. aureus*. It is worth noting that the high prevalence of *S. aureus* as carriers of the *bla*Z gene and the identification of methicillin-resistant strains among isolates from cheese pose a threat to the health of consumers of such products. It is important from a public health perspective to continuously monitor *S. aureus* in milk and milk products and their production environment. Continuous monitoring will allow the provision of measures for mitigating the public health threat associated with this resistant pathogen [55]. According to obtained results and findings showed by Ammar et al. [56], it is important to have comprehensive control measures during processing of handmade cheeses, and judicious application of antibiotics should be adopted to overcome the spread of ARM and minimalize the risk of human infection. 

The ability to form biofilm is one of the hazards encountered not only in healthcare facilities but also in the food industry. Pathogenic bacteria are able to form biofilms inside processing equipment, leading to food spoilage and subsequent health risks to consumers [57]. Biofilm formation by *Staphylococcus* spp. is an evolutionary advantage for this microorganism, as bacterial cells are resistant to adverse environmental conditions such as antimicrobial and sanitizing agents, desiccation, and UV radiation [58]. 

The obtained results showed that 59/93 (63.4%) of the isolates were able to produce biofilm. A similar percentage of strains capable of producing biofilm was shown in a study by Castro et al. (2020) [58], who classified 69.7% of isolates as biofilm-forming. The main factor responsible for biofilm production among *S. aureus* is considered to be PIA, which, together with other polymers such as teichoic acids, proteins, and extracellular DNA (eDNA), constitutes the main part of the extracellular matrix of biofilm-forming *S. aureus.* Genes contained in the *ica* locus are responsible for PIA biosynthesis and include N-acetylglucosamine transferase (*ica*A and *ica*D), PIA deacetylase (*ica*B), and PIA exporter (*ica*C) [59]. The presence of this operon was demonstrated among 26.9% of the tested isolates. According to the literature, in addition to the phenotypic ability to produce biofilm and the presence of the *ica* operon, this phenomenon may be related to the presence of other proteins (PIA-independent) that are independent of the *ica* operon [60]. In this regard, it is necessary to determine the occurrence of other genes that were responsible for biofilm formation and apply them to the study of biofilm formation on dairy farm equipment and utensils [29]. 

The current study found that the majority of examined isolates carried genes encoding for some of the studied enterotoxins. In the literature, other authors point to the co-occurrence of *sed* and *selj* genes due to the common location of these genes on the same plasmid [4,61], which was not confirmed by the results of the present study. However, Jorgensen et al. [62] showed that more than 50% of *S. aureus* strains isolated from cow’s milk and 14.7% isolated from different stages of cheese production possessed enterotoxin genes, but only *seg* and *sei* markers were identified.

## 4. Materials and Methods

### 4.1. Sample Collection

A total of 180 samples from various stages of artisanal cheese production from unpasteurized milk were collected from farms in the northeastern and southern parts of Poland. The samples were collected along the cheese production chain and included: raw milk, semi-finished products (heated milk, curd, and formed cheese), final products, and swabs from the production environment (from food and non-food contact surfaces).

The following samples were collected from cheese production stages: 18 of raw milk, 72 of semi-finished products (heated milk, n = 18; curd, n = 18; whey, n = 18; brine, n = 18), 18 of final products, and 72 swabs from the production environment, i.e., swabs from: tank (n = 18), jar (n = 18), form (n = 18), and sink (n = 18). Fluid and solid samples were collected at volumes of 100 mL or 100 g in sterile plastic tubes. Swab samples were collected with sterile cotton swabs (Equimed, Cracow, Poland) that were moistened in sterile peptone water, rolled on the test surface, and placed in tubes with transport media. The samples were collected in sterile containers and transported immediately to the microbiological laboratory.

### 4.2. Isolation and Identification of S. aureus Isolates

For isolation, Baird Parker agar (Merck Millipore, Darmstadt, Germany) was used. After 48 h of incubation at 37 ± 1 °C, one colony with typical morphology for *S. aureus* (black and shiny with a thin white border and surrounded by a light area) was selected for further identification. Firstly, suspected bacterial colonies were subjected to conventional methods such as Gram-stain, catalase test, and coagulase test.

Next, identification was performed using MALDI-TOF MS (Matrix-Assisted Laser Desorption/Ionization Time-of-Flight Mass Spectrometry) analysis with a Vitek MS instrument (bioMérieux, Marcy l’Etoile). Before the analysis, isolates were streaked on tryptic soy agar (TSA) (Merck, Darmstadt, Germany) and incubated at 37 ± 1 °C overnight. One colony was then applied to one spot of the test slide with an inoculating loop, covered with 1 μL of matrix solution (α-cyano-4-hydroxy- cinnamic acid) (bioMérieux, Marcy l’Etoile, France), and completely air-dried [63].

### 4.3. Antimicrobial Susceptibility Testing by Microdilution Broth Assay

Antimicrobial susceptibility profiles for the *S. aureus* isolates were determined by microbroth dilution in accordance with ISO 20776-1 “Susceptibility testing of infectious agents and evaluation of the performance of antimicrobial susceptibility test devices. Annex B: Solvents and diluents for making stock solutions of selected antimicrobial agents” [64] using 96-well bottom polystyrene plates. 

The study used amikacin, tobramycin clarithromycin, vancomycin (Sigma Aldrich, Darmstadt, Germany), tigecycline, tetracycline, linezolid, nitrofurantoin, oxacillin, benzylpenicillin, erythromycin, gentamycin, kanamycin, cefoxitin, azithromycin, quinupristin/dalfopristin, trimethoprim, trimethoprim/sulfamethoxazole, clindamycin, rifampicin, fusidic acid, and moxifloxacin, chloramphenicol (TOKU-E; Bellingham, WA, USA). Stock antibiotic solutions were prepared using recommended solvents and diluents according to ISO 20776-1, Annex B: “Solvents and diluents for making stock solutions of selected antimicrobial agents” [1]. On each day of testing, stock antibiotic solutions were diluted according to ISO 20776-1, Annex C: “Preparation of working dilutions of antimicrobial agents for use in broth dilution susceptibility tests” [1] to appropriate starting concentrations for serial dilution in the Mueller–Hinton broth. The concentration applied to the plate was twice the target concentration. *S. aureus* isolates were plated on TSA (Merck, Darmstadt, Germany) and incubated overnight. Next, single pure colonies from overnight culture were suspended in 0.9% saline solution to obtain turbidity equal to the McFarland 0.5 turbidity. Turbidity was checked using a McFarland densitometer DEN 1B, Bioscan (Bioscan, Riga, Latvia). Each 50 µL of antibiotic solutions and bacteria solutions was added to the well of each row. One set of wells was left blank as media controls and another as growth controls. The assay was performed in triplicate.

The 96-well plates (Promed^®^, Torreglia, Italy) were incubated at 35 ± 2 °C under aerobic conditions for 20 h. At the end of the incubation, the turbidity of the medium was observed. The lowest concentration at which no bacterial growth (medium turbidity) was observed is the MIC. Results were interpreted as susceptible, intermediate resistant, or resistant. Results were interpreted according to the criteria of the Clinical and Laboratory Standards Institute guidelines CLSI 2022 [65]. Results were read only when there was sufficient growth of the test organism (i.e., obvious button or definite turbidity in the positive growth control) and when there was no growth in the un-inoculated or negative growth control. The amount of growth in each well was compared with that in the positive growth control, and the MIC recorded was the lowest concentration of agent that completely inhibited visible growth. The antimicrobials, dilution ranges, and cut-off values used in the study are described in Table 5. *S. aureus* strain ATCC 25923 was used as a quality control.

The Multiple Antibiotic Resistance (MAR) index was calculated and interpreted as a/b, where “a” means the number of antibiotics to which the isolate was resistant, and “b” means the number of antibiotics for which the isolate was tested. The calculated formula is shown below [66]: MARindex=ab

### 4.4. In Vitro Biofilm Production Analysis

Biofilm production was tested using Congo red agar assay using the method described previously by Arciola [39]. The strains were cultured on TSA plates (Merck, Darmstadt, Germany) and supplemented with sucrose (Sigma Aldrich, Steinheim, Germany) and Congo red (Sigma Aldrich, Steinheim, Germany). Sucrose was added due to its characteristic of abundant exopolysaccharide synthesis among *Staphylococcus* spp. Based on colony appearance, strains were classified as slime-forming (black colonies) and non-slime-forming (bordeaux or pink colonies). Additionally, biofilm-producing ability was evaluating using the crystal violet assay method described previously by Kouidhi et al. [67].

### 4.5. Detection of Antibiotic Resistance, Enterotoxins, and Biofilm-Associated Genes among Isolates

The occurrence of tested genes was evaluated using a PCR assay with the specific primers and PCR reaction conditions as described in Appendix A. For all isolates, we tested for the presence of the following antibiotic resistance genes responsible for resistance to cefoxitin: *mec*A [68], and *mec*C [69]; tetracyclines: *tet*M, *tet*K, and *tet*L [70,71]; macrolides and lincosamides: *msr*(A/B), *erm*A, *erm*B, and *erm*C [72]; β-lactams: *bla*Z [73]; and aminoglycosides: *aac(6′)-Ie-aph(2″)-Ia* [74] and *aph(2″)-Ic* [75].

Genetic determinants responsible for the ability to form biofilm (*ica*ABCD, *eno*, *bap*, *agr*D, *sar*A, and *sig*B) were identified as described previously [76,77,78,79]. 

Among isolated strains, genes coding classical enterotoxins, enterotoxin-like toxins, exfoliative toxins (*eta* and *etd*), and shock syndrome toxins (*tstt-1*) were evaluated as described previously [80]. 

All PCR amplicons were electrophoresed on 1.5% (*w*/*v*) agarose (Agarose Basica LE) in 1× TBE (Tris-borate-EDTA) buffer stained with 0.5 μg/mL Midori green (ABO, Gdańsk, Poland) and visualized under ultraviolet light using the G-Box system (Syngene, Cambrige, UK).

### 4.6. Statistical Analyses

All statistical analyses were performed using GraphPad Prism software version 8.0 (GRAPH PAD Software Inc, San Diego, CA, USA). To determine the relationship between phenotypic and genetic determinants, the chi-square Pearson test was used. All correlation analyses were calculated using Pearson correlation. The occurrence of genes was marked as “1” when the gene was present, and “0” when it was absent, in which case the results of the Pearson correlation are identical to the point–biserial correlation. Results were considered statistically significant at *p* < 0.05. The R statistical platform (https://www.r-project.org) (accessed on 13 September 2022) and RStudio program (https://rstudio.com/) (accessed on 13 September 2022) were used for large-scale analysis and data visualization.

## 5. Conclusions

In this study, it was found that artisanal cheese chain production might potentially lead to contamination of finished cheeses with multidrug-resistant and virulent *S. aureus*. The application of good manufacturing practices and standard sanitation operating procedures in the farmhouse industry must be stricter. Occurrence of antibiotic-resistant strains in the food processing chain implies the need to monitor *S. aureus* to identify patterns of antimicrobial susceptibility to elucidate antibiotic-resistance transmission routes through the food chain. In addition, the presence of enterotoxin genes among *S. aureus* may be a health concern for consumers of artisanal cheeses. Our results showed that it is necessary to follow up with hygienic measures to prevent or minimize the contamination of handmade products. 

## Figures and Tables

**Figure 1 molecules-27-06569-f001:**
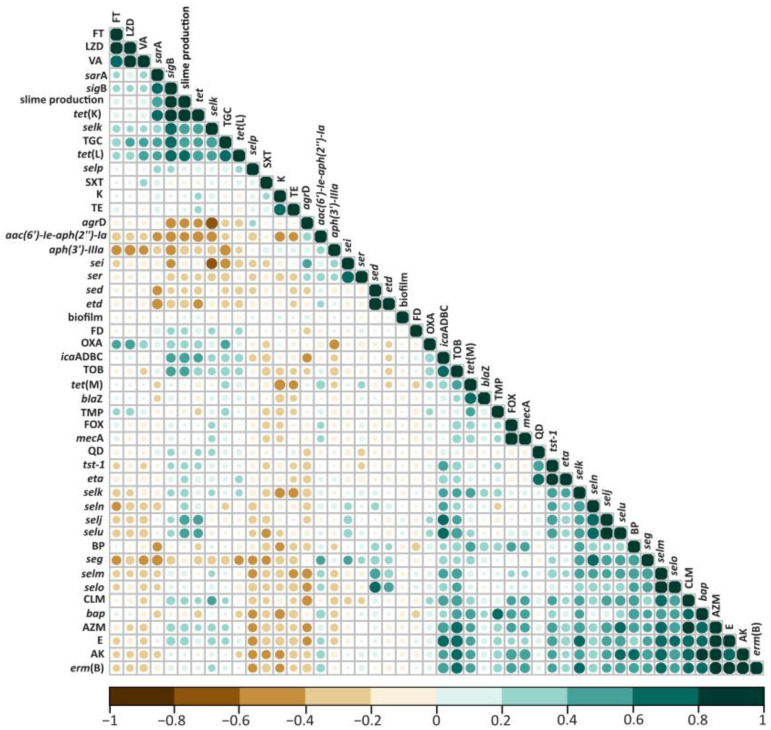
Correlation matrix indicating all correlations between pairs of genetic or phenotypic determinants (resistance phenotypes).

**Table 1 molecules-27-06569-t001:** Contamination with *S. aureus* at different stages of cheese production chain.

Source	No. of Samples	No. (%) of *S. aureus*-Positive Samples
**Sample**		
Raw milk	18	10 (55.6%)
Heated milk	18	13 (72.2%)
Curd	18	15 (83.3%)
Whey	18	8 (44.4%)
Brine	18	10 (55.6%)
Cheese	18	10 (55.6%)
**Swab**		
Tank	18	8 (44.4%)
Jar	18	8 (44.4%)
Form	18	9 (50.0%)
Sink	18	2 (11.1%)
**Total**	180	93 (51.7%)

**Table 2 molecules-27-06569-t002:** Antimicrobial resistance profiles of *S. aureus* against tested antimicrobial agents.

		Samples						Swabs				
Antimicrobial Class	Antimicrobial Agent	Raw Milk (n = 9)	Heated Milk (n = 14)	Curd (n = 15)	Whey (n = 8)	Brine (n = 10)	Cheese (n = 10)	Tank (n = 8)	Jar (n = 8)	Form (n = 9)	Sink (n = 2)	Total (n = 93)
**Aminoglycosides**	**AK**	0	4 (28.6%)	6 (40.0%)	4 (50.0%)	0	3 (30.0%)	3 (37.5%)	1 (12.5%)	2 (22.2%)	0	**23 (24.7%)**
	**TOB**	4 (44.4%)	6 (42.9%)	5 (33.3%)	2 (25.0%)	2 (20.0%)	4 (40.0%)	3 (37.5%)	3 (37.5%)	2 (22.2%)	1 (50.0%)	**32 (34.4%)**
	**K**	1 (11.1%)	0	1 (6.7%)	0	1 (10.0%)	1 (10.0%)	1 (12.5%)	0	0	1 (50.0%)	**6 (6.5%)**
**Cephalosporins**	**FOX**	1 (11.1%)	1 (7.1%)	3 (20.0%)	1 (12.5%)	0	2 (20.0%)	1 (12.5%)	1 (12.5%)	2 (22.2%)	0	**12 (12.9%)**
**Macrolides, lincosamides and** **streptogramins**	**AZM**	0	3 (21.4%)	3 (20.0%)	2 (25.0%)	0	3 (30.0%)	1 (12.5%)	3 (37.5%)	2 (22.2%)	0	**17 (18.3%)**
	**CLM**	2 (22.2%)	3 (21.4%)	2 (13.3%)	2 (25.0%)	0	3 (30.0%)	0	0	3 (33.3%)	0	**15 (16.1%)**
	**E**	2 (22.2%)	5 (35.7%)	2 (13.3%)	2 (25.0%)	0	3 (30.0%)	2 (25.0%)	3 (37.5%)	2 (22.2%)	0	**21 (22.6%)**
**β-lactams**	**BP**	5 (55.6%)	8 (57.1%)	11 (73.3%)	7 (87.5%)	5 (50.0%)	6 (60.0%)	4 (50.0%)	4 (50.0%)	4 (44.4%)	0	**54 (58.1%)**
	**OXA**	0(0.0%)	1 (7.1%)	1 (6.7%)	1 (10.0%)	1 (10.0%)	3 (30.0%)	1(12.5%)	0	1 (11.1%)	0	**9 (9.7%)**
**Oxazolidinones**	**LZD**	0	0	0	0	0	1 (10.0%)	0	0	0	0	**1 (1.1%)**
**Nitrofurantoins**	**FT**	1 (11.1%)	0	0	0	0	1 (10.0%)	0	0	0	0	**2 (2.2%)**
**Tetracyclines**	**TE**	2 (22.2%)	5 (35.7%)	2 (13.3%)	0	4 (40.0%)	2 (20.0%)	2 (25.0%)	2(25.0%)	0	0	**19 (20.4%)**
	**TGC**	2 (20.0%)	3 (21.4%)	4 (26.7%)	2 (25.0%)	1 (10.0%)	3 (30.0%)	2 (25.0%)	3 (37.5%)	2 (22.2%)	0	**22 (23.7%)**
**Sulfonamides**	**TMP**	3 (33.3%)	2 (14.3%)	1 (6.7%)	2 (25.0%)	0	2 (20.0%)	1 (12.5%)	2 (25.0%)	2 (22.2%)	0	**15 (16.1%)**
	**SXT**	1 (11.1%)	3 (21.4%)	3 (20.0%)	0	1 (10.0%)	2 (20.0%)	1 (12.5%)	0	0	0	**11 (11.8%)**
**Glycopeptides**	**VA**	0	0	1 (6.7%)	0	0	1 (10.0%)	0	0	0	0	**2 (2.2%)**
**Streptogramins**	**QD**	0	0	0	0	1 (10.0%)	0	0	0	1 (11.1%)	0	**2 (2.2%)**
**Steroidal**	**FD**	1 (11.1%)	0	0	0	0	0	0	0	0	1 (50.0%)	**2 (2.2%)**

Abbreviations: n—number of *S. aureus* isolates; AK—amikacin; TOB—tobramycin; K—kanamycin; FOX—cefoxitin; AZM—azithromycin; CLM—clarithromycin; E—erythromycin; BP—benzylpenicillin; OXA—oxacillin; LZD—linezolid; FT—nitrofurantoin; TE—tetracycline; TGC—tigecycline; TMP—trimethoprim; SXT—trimethoprim/sulfamethoxazole; VA—vancomycin; QD—quinupristin/dalfopristin; FD—fusidic acid.

**Table 3 molecules-27-06569-t003:** Occurrence of antimicrobial resistance genes against tested antimicrobial agents in *S. aureus*.

		Samples						Swabs				
Antimicrobial Class	Antibiotic Resistance Gene	Raw Milk (n = 9)	Heated Milk (n = 14)	Curd (n = 15)	Whey (n = 8)	Brine (n = 10)	Cheese (n = 10)	tank (n = 8)	Jar (n = 8)	Form (n = 9)	Sink (n = 2)	Total (n = 93)
**Oxacillin**	*mec*A	1 (11.1%)	1 (7.1%)	3 (20.0%)	1 (12.5%)	0	2 (20.0%)	1 (12.5%)	1 (12.5%)	2 (22.2%)	0	**12 (12.9%)**
**Penicillin**	*bla*Z	9 (100%)	13 (92.9%)	15 (100.0%)	8 (100.0%)	8 (80.0%)	10 (100.0%)	8 (100.0%)	8 (100.0%)	9 (100.0%)	1 (50.0%)	**89 (95.7%)**
**Tetracyclines**	*tet*K	5 (55.6%)	9 (64.3%)	10 (66.7%)	4 (50.0%)	5 (50.0%)	7 (70.0%)	5 (62.5%)	6 (75.0%)	6 (66.7%)	2 (100%)	**59 (63.4%)**
	*tet*M	6 (66.7%)	12 (85.7%)	14 (93.3%)	6 (75.0%)	5 (50.0%)	9 (90.0%)	7 (87.5%)	7 (87.5%)	9 (100.0%)	1 (50.0%)	**76 (81.7%)**
	*tet*L	1 (11.1%)	1 (7.1%)	3 (20.0%)	2 (25.0%)	1 (10.0%)	2 (20.0%)	1 (12.5%)	2 (25.0%)	2 (22.2%)	1 (50.0%)	**16 (17.2%)**
**Macrolides, lincosamides, and** **streptogramins**	*erm*B	2 (22.2%)	5 (35.7%)	5 (33.3%)	3 (37.5%)	1 (10.0%)	2 (20.0%)	3 (37.5%)	2 (25.0%)	3 (33.3%)	1 (50.0%)	**27 (29.0%)**
**Aminoglycosides**	*aac(6′)-Ie-aph(2″)-Ia*	3 (33.3%)	9 (64.3%)	8 (53.3%)	3 (37.5%)	8 (80.0%)	6 (60.0%)	8 (100.0%)	4 (50%)	7 (77.8%)	1 (50.0%)	**56 (60.2%)**
	*aph(3′)IIIa*	9 (100.0%)	12 (85.7%)	12 (80.0%)	8 (100.0%)	10 (100.0%)	9 (90.0%)	8 (100.0%)	7 (87.5%)	9 (100.0%)	2 (100%)	**86 (92.5%)**

n—number of *S. aureus* isolates.

**Table 4 molecules-27-06569-t004:** Virulence characteristics among *S. aureus* isolates.

		Samples						Swabs				
		Raw Milk (n = 9)	Heated Milk (n = 14)	Curd (n = 15)	Whey (n = 8)	Brine (n = 10)	Cheese (n = 10)	Tank (n = 8)	Jar (n = 8)	Form (n = 9)	Sink (n = 2)	Total (n = 93)
**Biofilm MTP method**	strong	5 (55.6%)	9 (64.3%)	12 (80.0%)	4 (50.0%)	7 (70.0%)	9 (90.0%)	3 (37.5%)	3 (37.5%)	4 (44.4%)	2 (100%)	**58 (62.4%)**
	intermediate	1 (11.1%)	1 (7.1%)	-	-	-	-	-	-	-	-	**2 (2.2%)**
	weak	3 (33.3%)	4 (28.6%)	3 (20.0%)	4 (50.0%)	3 (30.0%)	1 (10.0%)	5 (62.5%)	5 (62.5%)	5 (55.6%)		**33 (35.4%)**
**Slime production**		4 (44.4%)	7 (50.0%)	9 (60.0%)	4 (50.0%)	4 (40.0%)	7 (70.0%)	3 (37.5%)	5 (62.5%)	3 (33.3%)	2 (100%)	**48 (51.6%)**
**Biofilm-associated genes**	*ica*ADBC	2 (22.2%)	4 (28.6%)	4 (26.7%)	3 (37.5%)	1 (10.0%)	4 (40.0%)	2 (25.0%)	2 (25.0%)	2 (22.2%)	1 (50.0%)	**25 (26.9%)**
	*bap*	3 (33.3%)	7 (50.0%)	8 (53.3%)	2 (25.0%)	1 (10.0%)	5 (50.0%)	2 (25.0%)	1 (12.5%)	3 (33.3%)	0	**32 (34.4%)**
	*eno*	9 (100%)	14 (100.0%)	15 (100.0%)	8 (100.0%)	10 (100%)	10 (100.0%)	8 (100.0%)	8 (100.0%)	9 (100.0%)	2 (100%)	**93 (100.0%)**
	*sig*B	5 (55.6%)	6 (42.9%)	8 (53.3%)	4 (50.0%)	2 (20.0%)	6 (60.0%)	3 (37.5%)	5 (62.5%)	7 (77.8%)	1 (50.0%)	**42 (45.2%)**
	*sar*A	7 (77.8%)	13 (92.9%)	12 (80.0%)	6 (75.0%)	8 (80.0%)	8 (80.0%)	6 (75.0%)	5 (62.5%)	7 (77.8%)	2 (100%)	**74 (79.6%)**
	*agr*D	2 (22.2%)	6 (42.9%)	5 (33.3%)	4 (50.0%)	5 (50.0%)	3 (30.0%)	2 (25.0%)	3 (37.5%)	3 (33.3%)	-	**33 (35.5%)**
**Enterotoxigenic genes**	*sed*	-	1 (7.1%)	-	-	1 (10.0%)	1 (10.0%)	-	-	-	-	**3 (3.2%)**
	*seg*	4 (44.4%)	9 (64.3%)	8 (53.3%)	7 (87.5%)	5 (50.0%)	6 (60.0%)	5 (62.5%)	4 (50.0%)	5 (55.6%)	1 (50.0%)	**54 (58.1%)**
	*sei*	2 (22.2%)	4 (28.6%)	1 (6.7%)	1 (12.5%)	-	-	2 (25.0%)	2 (25.0%)	1 (11.1%)	-	**13 (14.0%)**
	*selj*	1 (11.1%)	3 (37.5%)	4 (26.7%)	3 (37.5%)	2 (20.0%)	4 (40.0%)	2 (25.0%)	2 (25.0%)	3 (33.3%)	-	**26 (28.0%)**
	*selk*	7 (77.8%)	10 (71.4%)	10 (66.7%)	4 (50.0%)	4 (40.0%)	7 (70.0%)	3 (37.5%)	5 (62.5%)	5 (55.6%)	2 (100.0%)	**57 (61.3%)**
	*selm*	-	2 (14.3%)	3 (20.0%)	2 (25.0%)	3 (30.0%)	3 (30.0%)	3 (37.5%)	-	1 (11.1%)	1 (50.0%)	**18 (19.4%)**
	*seln*	2 (22.2%)	7 (50.0%)	4 (26.7%)	6 (75.0%)	4 (40.0%)	6 (60.0%)	4 (40.0%)	3 (37.5%)	5 (55.6%)	1 (50.0%)	**42 (45.2%)**
	*selo*	-	1 (7.1%)	3 (20.0%)	1 (12.5%)	2 (20.0%)	3 (30.0%)	3 (37.5%)	-	-	1 (50.0%)	**14 (15.1%)**
	*selp*	2 (22.2%)	-	4 (26.7%)	2 (25.0%)	3 (30.0%)	1 (10.0%)	-	3 (37.5%)	1 (11.1%)	-	**16 (17.2%)**
	*ser*	1 (11.1%)	4 (28.6%)	3 (20.0%)	1 (12.5%)	-	-	2 (25.0%)	1 (12.5%)	1 (11.1%)	1 (50.0%)	**14 (15.1%)**
	*selq*	3 (33.3%)	8 (57.1%)	8 (53.3%)	6 (75.0%)	2 (20.0%)	6 (60.0%)	3 (37.5%)	3 (37.5%)	3 (33.3%)	1 (50.0%)	**43 (46.2%)**
	*selu*		5 (35.7%)	5 (33.3%)	3 (37.5%)	2 (20.0%)	4 (40.0%)	3 (37.5%)	2 (25.0%)	4 (44.4%)	-	**29 (31.2%)**
	*eta*	-	1 (7.1%)	-	-	-	-	-	1 (12.5%)	1 (11.1%)	-	**3 (3.2%)**
	*etd*	-	-	-	-	1 (10.0%)	1 (10.0%)	-	-	-	-	**2 (2.2%)**
	*tst-1*	-	-			-	1 (10.0%)		1 (12.5%)	2 (22.2%)	-	**4 (4.3%)**

(-) not detected. n—number of *S. aureus* isolates.

**Table 5 molecules-27-06569-t005:** Antimicrobials, dilution ranges, and cut-off values used for minimum inhibitory concentration (MIC) determination in *S. aureus*.

Antimicrobial Class	Antimicrobials Agent	Dilution Range (mg/L)	Cut-Off Values (mg/L); Resistant > R
**Aminoglycosides**	Amikacin (AK)	0.25–128	16
	Tobramycin (TOB)	0.063–16	2
	Gentamycin (CN)	1–4	2
	Kanamycin (K)	0.25–128	16
**Cephalosporins**	Cefoxitin (FOX)	1–32	4
**Macrolides**	Azithromycin (AZM)	0.25–16	2
	Clarithromycin (CLM)	0.063–16	4
	Erythromycin (E)	0.125–16	1
**β-lactams**	Benzylpenicillin (BP)	0.063–8	0.125
	Oxacillin (OXA)	0.063–16	2
**Oxazolidinones**	Linezolid (LZD)	0.5–16	4
**Nitrofurantoins**	Nitrofurantoin (FT)	2–128	64
**Tetracyclines**	Tetracycline (TE)	0.063–32	2
	Tigecycline (TGC)	0.016–8	0.5
**Sulfonamides**	Trimethoprim (TMP)	0.125–32	2
	Trimethoprim/sulfamethoxazole (SXT)	0.125:2.375–32:608	2
**Glycopeptides**	Vancomycin (VA)	0.25–64	2
**Streptogramins**	Quinupristin/dalfopristin (QD)	0.125–8	2
**Lincosamides**	Clindamycin (DA)	0.031–32	2
**Rifamycins**	Rifampicin (RD)	0.002–8	0.06
**Steroidal**	Fusidic acid (FD)	0.5–4	1
**Fluoroquinolones**	Moxifloxacin (MXF)	0.004–8	0.25
**Phenicols**	Chloramphenicol (C)	2–32	8

## Data Availability

The dataset used during this study is available from the author upon reasonable request.

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
