# Peer review of "Occurrence and Characteristics of Staphylococcus aureus Strains along the Production Chain of Raw Milk Cheeses in Poland"

_molecules, 2022, doi:10.3390/molecules27196569_

Round 1
Reviewer 1 Report
Dear Authors
Congratulations on a successful and well-written job. I found little things to improve and lighten your work. Notes below.
1 1. In the antagonistic test in the turbidity method according to SO 20776-1 “Susceptibility testing of infectious agents and evaluation of the performance of antimicrobial susceptibility test devices” Mueller-Hinton medium is used which is not clear. How did the authors incorporate this element in the absorbance reading?
22. Line 272: Citing the work of Avila-Novoa et al. 2021 is not correct. Cite the original publication “Presence of icaA and icaD genes and slime production in a collection of staphylococcal strains from catheter-associated infections” C.R. Arciola, L. Baldassarri and L. Montanaro, Journal of Clinical Microbiology, 39 (2001), pp. 2151-2156.
33. It should also be added that the key factor in this medium is the addition of sucrose as an inducer of exopolysaccharide synthesis by Staphylococcus strains.
Reviewer 2 Report
This is an interesting study on the distribution of antibiotic resistance and virulence genes in Staphylococcus aureus isolated from milk products and processing facilities in Poland.
My comments and remarks are given below.
Abstract.
Line 15. “Frequently” is misleading. Omit or add information about the targeted genes.
Line 17. Do the authors mean “isolates, the resistance to macrolides was shown”?
Line 19. Phenotypic resistance or phenotypically resistant.
Line 25. Replace “resistant” with “resistance”, “this” with “our”.
Keywords: SE genes – use full names for abbreviations at first mentioning
Introduction
Line 45-47. Not clear how the finished product may be contaminated during different production stages?
Line 48-52. Why do the authors describe MRSA-related conditions if the aim of the present study is to investigate the presence and characteristics of S. aureus?
To state the problem, the authors need to clearly distinguish between the infections or transmissions associated with S. aureus. As example, the present text could be read as S. aureus may cause opportunistic mastitis in humans? There are also different types of transmission: farms vs. food chain.
Line 53-57. How representative is the study conducted in Brazil in terms of global surveillance and problems of all continents and all countries?
Line 58. “Another virulence factor”. Which was the first? In abstract, the authors distinguish between the AMR and virulence gene factors.
Authors may add a short description of importance of biofilms and responsible genes in S. aureus isolates.
Results.
Line 77. What is CPS?
Were any multiresistant isolates revealed?
Line 94-97. Please add the total no. of isolates were tested.
For 2.2. subsection the authors need to specify a total no. of isolates were tested for AMR resistance and detection of AMR genes.
For all tables, n=.. are not explained/ Are they no. of samples or positive samples or number of isolates?
Line 115. Table 4?
Line 117. Was this association determined statistically? Please add p-value if yes.
Authors should provide some interesting link to the present results, eg.
Were associations between AMR, virulence genes and biofilm formation identified?
Were association between the phenotypical AMR characteristics and the presence of AMR genes found?
Were the differences in AMR, virulence, biofilm development activity pattern between products and surfaces identified?
Otherwise it is difficult establish the differences or similarities in resistance and virulence of S. aureus as has been stated in the title of the present study.
Line 143. Add the appropriate reference after “milk”.
Line 144. Add the appropriate reference after “products”.
Line 148. Is the presence in the present study has been compared with other results? Please add the numbers of investigated samples, line 150.
Line 153. What do the author mean with “low quality”?
Line 155-157. How this sentence is related to the present study?
Line 161. Please add more that one reference to support multiple reports on resistance to beta-lactams.
Line 165. Please add the number of investigated/positive samples.
Line 172. Please provide a number of MDR isolates.
Line 204- 206. Where are those many authors?
Line 208. What are SE genes?
Materials and methods.
Line 214. Authors should list “various stages” of cheese production.
Please also describe the sampling technique and sampling plan.
Line 221. CPS?
Line 223. Please describe the typical morphological characteristics for selected colonies?
Line 223. CoPS?
How many colonies per agar were selected for confirmation?
Conclusions.
Line 287. “contamination” of what?
Line 289-292. How the present study may contribute the cheese processing technologies?
Conclusions should be more focused on the study content.
Reviewer 3 Report
The authors addressed the occurrence of S. aureus in artisanal cheese production chain from unpasteurized milk. Moreover, they investigated the antibiotic resistance, enterotoxicity, the ability to biofilm formation and slime production as well as frequently virulence-related genes among S. aureus strains. It is greatly suggested that the manuscript is not ready to be accepted now. I have several comments listed below.
1. Rationale:
· The tile is not constructed well. It must be rewritten.
· The abstract lacks the materials and methods used.
· The results of resistance in the abstract are needed to be rephrased to be clearer.
· Introduction must be focus on the problem the research dealt with and how will the authors solve this problem in addition to determining the gap in this point in previous researches.
· The authors must mention the details of the samples collected and the number of samples per each type.
· The authors did not identify the isolates first before any further analysis at the molecular level. This must be included in the manuscript to make the results more reliable.
· In table 2, specify the first column and the second column must be antimicrobial agent.
· What is the impact of the data of the samples' types on the prevalence of resistance or virulence?.
· The authors must illustrate well the best definition for the isolates to be MDR and also must determine the percentages of MDR isolates in the study.
· The authors should focus on the correlation between the phenotypic and genotypic results for resistance and biofilm detection with referring to the r values.
· The discussion must include the causes for the similarities or differences for all items not only mention what are different or similar than other researches.
2. Logics:
· The titles of materials and methods sections must be specified more.
· Is the primary isolation medium for S. aureus CPS Baird Parker agar with rabbit plasma fibrinogen?. The primary selective media for isolation of S. aureus is mannitol salt agar or Baird Parker agar media.
· In antimicrobial susceptibility testing in materials section, where are the preparations of bacterial suspensions and antibiotic stock solutions?
· What is the reference for calculating the multiple antibiotic resistance index?.
· Phenotypic biofilm production must be before genetic detection of biofilm genes.
· What is the impact of the authors’ findings as methicillin resistance is now widely spread and the vancomycin is more dangerous than methicillin resistance. The authors must define the isolates as MRSA phenotypically and genotypically and illustrate the results in relation to MRSA after its confirmation and also VRSA if present.
· The authors must state the references for all methods they stated.
· All reagents, media, solutions and equipment in the materials section must be supplied with the company and country names.
· Where is the ethical statement for the study?
· The authors must focus on their important findings in the conclusion section.
3. Typo issues:
· Antimicrobials or their classes must be mentioned by capital letters if they are mentioned in the beginning of the sentence. Moreover, all words in the tables' cells must be mentioned by capital letters.
· There were some errors in the structure of several sentences.
· Under all table s, all abbreviations must be mentioned as footnotes.
· The authors must write what each abbreviated word stands for before using the abbreviation for the first time.
Finally, the following references are very important for the topic of the manuscript. Please, refer to them in the manuscript:
- Genetic basis of resistance waves among methicillin resistant Staphylococcus aureus isolates recovered from milk and meat products in Egypt.
- Clonal diversity and epidemiological characteristics of ST239-MRSA strains.
- What is behind phylogenetic analysis of hospital‐, communityand livestock‐associated methicillin‐resistant Staphylococcus aureus?.
Round 2
Reviewer 2 Report
All my comments and suggestions had been carefully addressed.
Author Response
Dear Reviewer 2,
Once again, thank you very much for taking the time to review our manuscript, which has improved the quality.
Reviewer 3 Report
Dear authors,
Please, find the attached PDF file.
